# Learning Skill-level Student Abilities with Item Response Theory

## Abstract

Knowledge tracing (KT) aims to estimate knowledge states of students over a given set of skills based on their historical learning activities. The learned knowledge states of students can be used to build skill-meters to understand the weak areas of students so that proper interventions can be taken to help students. Many deep learning models have been applied to KT with encouraging performance, but they either have relatively low accuracy or do not directly generate students' knowledge states at skill level for skill-meter building. Item Response Theory (IRT) models student knowledge states (ability) and question characteristics separately. A question arising naturally is whether we can use IRT to estimate students' knowledge states at skill level while achieving high prediction accuracy at the same time. We examined existing IRT based deep KT models and found that none of them achieves this objective. Most existing IRT-based models either learn overall student abilities or question-level student abilities. Overall student abilities are too summative, and it is hard to tell the weak areas of students from a single value. Question-level abilities are too fine-grained. When there are a large number of unique questions per skill, they can cause information overload for teachers. In this paper, we propose an IRT-based deep KT model called SKKT-IRT to learn skill-level student abilities which provide just the right amount of information for teachers to understand students' knowledge states. Our model consists of an LSTM layer to learn student historical states, a student ability network for learning skill-level student abilities, a question difficulty network for learning question difficulties and a question discrimination network for learning question discrimination. It also learns question-skill relationships as an auxiliary task so that the embedding of a skill can better capture the information of its questions. We further regularize the outputs of question difficulty network and question discrimination network for better performance. Our experimental results show that our model achieves the objective of learning skill-level student abilities with SOTA accuracy. It is also very efficient and produces consistent outputs to be easily used for downstream tasks like adaptive learning and personalized recommendations.

## 1 Introduction

Knowledge tracing (KT) is a key component in intelligent tutoring systems (ITSs) for personalized and adaptive learning. It aims to estimate knowledge states of students over a set of skills based on students' historical learning activities. Given that the ground-truth knowledge states of students over skills are usually unknown, the performance of knowledge tracing models is usually assessed using the next question correctness prediction task. Let $x = (u, q, y)$ be a learning activity of a student, where $u$ is a student ID, $q$ is a question ID, and $y$ is a binary variable (class label) indicating whether student $u$ answered question $q$ correctly or not. Each question has one or more skills associated with it. The next question correctness prediction task can be formulated as follows: given a sequence $S_u = \langle x_1, x_2, \cdots, x_t \rangle$ containing historical learning activities of a student $u$, predict whether student $u$ can answer the next question at $t+1$ correctly.

The knowledge states learned by knowledge tracing models can be used to build skill-meters (Corbett & Anderson, 1994) of students. Students can use the skill-meters to understand how well they master each skill. Teachers can use the skill-meters to identify common weak skills in their classes. An example skill-meter of a student over eight skills is shown in Figure 1(a). It shows that the

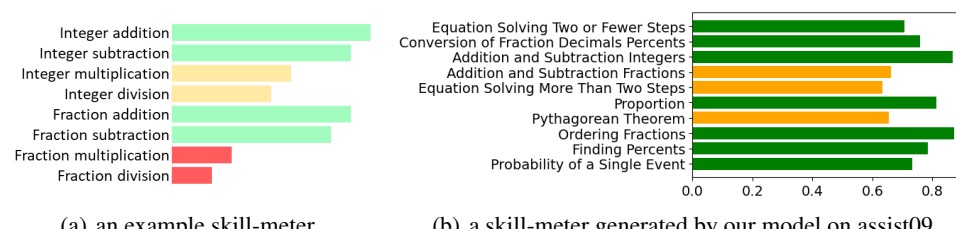

(a) an example skill-meter          (b) a skill-meter generated by our model on assist09

Figure 1: Example skill-meters.

student has mastered addition and subtraction very well, achieves some level of mastery on integer multiplication and division, and is doing poorly on fraction multiplication and division. The student may need to work more on integer multiplication and division before moving on to fraction multiplication and division. Early knowledge tracing models like Bayesian Knowledge Tracing (Corbett & Anderson, 1994, BKT) and Deep Knowledge Tracing (Piech et al., 2015, DKT) use skill information only to trace knowledge states at skill level. However, their prediction accuracy is low because important question information is not utilized. Many later models incorporate question information with much improved accuracy, but their predictions are thus on specific questions. Constructing skill-meters from predictions on individual questions is non-trivial.

Item Response Theory (IRT) (Lordn, 1980) models student knowledge states (ability) and question characteristics separately. More specifically, it models the probability of a student answering a question correctly as a logistic function of student ability (knowledge states) and question characteristics. A question arising naturally is whether we can use IRT to estimate students' knowledge states at skill level while achieving high prediction accuracy on the next question correctness prediction task at the same time. We examined existing IRT-based deep KT models and found that none of them achieves this objective. The first IRT-based deep KT model Deep-IRT (Yeung, 2019) learns overall student abilities using a key-value memory network. Overall ability is not informative enough as students may have different abilities over different skills. The accuracy of Deep-IRT is also much lower than SOTA. PKT (Sun et al., 2024a) and MIKT (Sun et al., 2024b) use the embedding of the next question at $t + 1$ together with the hidden representation of student learning history to generate student ability at $t + 1$. The student ability learned by them are thus at question level, which is too fine grained especially when the number of questions is large. DKT-IRT (Converse et al., 2021) is the only IRT-based deep KT model which learns skill-level student abilities, but its accuracy is very low, i.e., close to that of DKT (Piech et al., 2015).

We also found that existing IRT-based deep KT models may produce contradicting outputs. The question discrimination parameter learned by DKT-IRT can be negative while it should always be positive. For a question with negative discrimination, the probability of answering it correctly decreases with increased student ability as shown in Figure 2(a), which contradicts both IRT and common sense. MIKT learns question-level student abilities which may not always be consistent with learned question difficulties. At a given time point, we can use MIKT to estimate a student's question-level abilities over all questions. Figure 2(b) shows the difficulties of a set of questions from the same skill (x-axis) and abilities of a student over these questions (y-axis) estimated by MIKT at a given time point. The ability of the student can be higher on a harder question than that on an easier question with the same skill, which also contradicts IRT. End users will find it hard to trust and use these contradicting outputs for downstream tasks such as adaptive learning and personalized recommendations. Our model is able to eliminate the inconsistency caused by question-level student abilities. Our model learns skill-level student abilities, so the student has the same ability on questions from the same skill, and the probability of answering these questions correctly decreases with the increased question difficulty as shown in Figure 2(c).

In this paper, we propose an IRT-based deep KT model which learns skill-level student abilities (knowledge states) without sacrificing accuracy or consistency. We design our model architecture carefully to achieve this objective. Our model uses an LSTM sublayer to generate hidden representations of student history sequences, a student ability network to map outputs of the LSTM sublayer and skill embeddings to student abilities over skills, a question difficulty network to transform question embeddings to question difficulties, and a question discrimination network to transform ques-

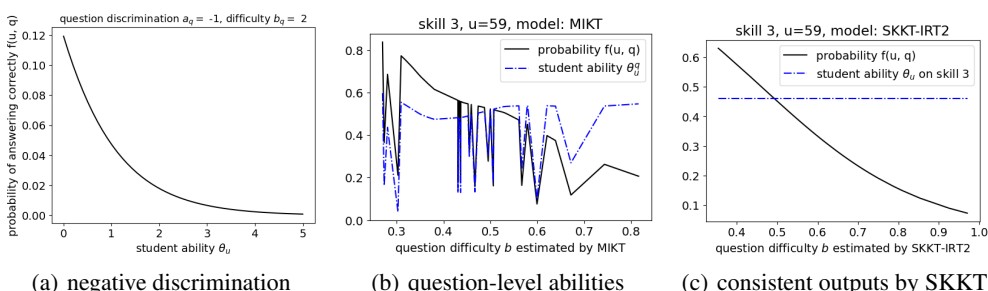

(a) negative discrimination     (b) question-level abilities     (c) consistent outputs by SKKT

Figure 2: (a) contradicting outputs caused by negative question discrimination by DKT-IRT: probability of answering correctly decreases with increased student ability; (b) contradicting outputs caused by question-level abilities by MIKT: question-level ability of a student can be higher on harder questions than on easier questions from the same skill; (c) consistent outputs by our model: probability of answering correctly decreases with increased question difficulties.

tion embeddings to question discrimination parameters. In addition, our model learns question-skill relationships as an auxiliary task so that the embedding of a skill can better capture its question information. We also regularize learned question difficulty and discrimination parameters to further improve model performance: loss penalty is imposed if learned question difficulties deviate from their statistics estimated from data, and if learned question discrimination parameters deviate from their default value of 1. The outputs of our model can be used easily for skill-meter building and other downstream tasks. Figure 1(b) shows a skill-meter built from skill-level student abilities learned by our model on assist09. The main contributions of our paper are summarized below:

- We propose an IRT-based deep KT model which learns skill-level student abilities with SOTA accuracy on the next question correctness prediction task. To the best of our knowledge, our work is the first IRT-based deep KT model achieving this objective.

- Existing IRT-based deep KT models may produce inconsistent outputs. We clearly point out the requirements of IRT and design our model architecture carefully so that all the requirements of IRT are satisfied and our model produces consistent outputs by design.

- We employ a number of techniques to improve the performance of our model, including an auxiliary task to learn question-skill relationships and two regularization terms to regularize the outputs of question difficulty network and question discrimination network.

- Our model supports both one-parameter item response function (1P-IRF) and two-parameter item response function (2P-IRF). We are the first to compare 1P-IRF and 2P-IRF under the same framework.

- Our model is very efficient. In particular, it is 50+ times faster than MIKT, which is the best performing IRT-based deep KT model in terms of accuracy.

- We conducted extensive experiments to show the performance of our model. Besides accuracy and efficiency, we also show that the IRT parameters generated by our model satisfy all the requirements of IRT and the question difficulties generated by our model have higher correlations with question difficulties estimated from data than existing IRT-based models

The rest of the paper is organized as follows. Section 2 introduces related work. Section 3 describes item response theory and its requirements. Our proposed model is presented in Section 4. Experiment results are reported in Section 5. Finally, Section 6 concludes the paper.

## 2 RELATED WORK

The knowledge tracing problem was first studied in (Corbett & Anderson, 1994), and a Bayesian Knowledge Tracing (BKT) model was proposed to model knowledge states of students using a Hidden Markov Model. Many different approaches have been developed since then, including further improvements to BKT (de Baker et al., 2008; Pardos & Heffernan, 2010; 2011; Yudelson et al.,

2013; Khajah et al., 2016), factor analysis models (Cen et al., 2006; 2008; Pavlik et al., 2009; Lindsey et al., 2014; Lan et al., 2014b;a; Vie & Kashima, 2019; Choffin et al., 2019) and deep KT models. For a review of these algorithms, please refer to (Liu et al., 2021b; Abdelrahman et al., 2023).

DKT (Piech et al., 2015) is the first algorithm using a deep learning model for knowledge tracing, and it uses LSTM. It takes skill IDs and student responses as inputs. Many more deep learning models are applied to knowledge tracing since then, including variants of RNN models (Yeung & Yeung, 2018; Minn et al., 2018; Wang et al., 2019a;b; Lee & Yeung, 2019; Liu et al., 2020; 2021a; Sun et al., 2022; Chen et al., 2023; Liu et al., 2023a), memory-augmented NN (Zhang et al., 2017; Abdelrahman & Wang, 2019), ConvNN (Shen et al., 2020), Graph NN (Nakagawa et al., 2019; Yang et al., 2020; Tong et al., 2020; Zhang et al., 2021), attentive models (Pandey & Karypis, 2019; Ghosh et al., 2020; Pandey & Srivastava, 2020; Choi et al., 2020; Shin et al., 2021; Huang et al., 2021; Lee et al., 2022; Yin et al., 2023; Wang et al., 2023; Huang et al., 2023) and hybrid models (Sun et al., 2024b; Ma et al., 2024). These deep learning based knowledge tracing models either have low accuracy or generate predictions at question-level only.

Several works use IRT for better interpretability. Deep-IRT (Yeung, 2019) transforms outputs of a key-value memory network to student overall abilities and uses a difficulty network to convert question embeddings to question difficulties, and then combines the two using a variant of 1P-IRF. DKT-IRT (Converse et al., 2021) is built from the DKT model, and it uses a variant of 2P-IRF in its prediction layer. The question discrimination parameters learned by DKT-IRT can be negative, which contradicts IRT. PKT (Sun et al., 2024a) and MIKT (Sun et al., 2024b) learn question-level student abilities, which are too fine grained especially when the number of questions is large. PKT (Sun et al., 2024a) uses 2P-IRF and it unnecessarily restricts the values of question discrimination $a_q$ to be within $[0, 1]$ while in reality, $a_q$ can be larger than 1.

## 3 ITEM RESPONSE THEORY

Item response theory (IRT) (Lordn, 1980) is a framework in psychometrics used for explaining the relationship between latent traits (e.g., student ability) and their manifestations (e.g. student responses to questions). It models the probability of a student answering a question correctly as a logistic function of student ability and question characteristics. The one-parameter item response function (1P-IRF), also called Rasch model, uses only one parameter for questions. It calculates the probability of a student $u$ answering a question $q$ correctly as follows, where $\theta_u$ is the ability of student $u$ and $b_q$ is the difficulty of question $q$.

$$f(u, q) = \sigma(\theta_u - b_q) = \frac{1}{1 + e^{-(\theta_u - b_q)}} \tag{1}$$

The two-parameter item response function (2P-IRF) also considers question discrimination, and it calculates the probability of a student $u$ answering a question $q$ correctly as follows, where $a_q$ is the discrimination of question $q$, and it controls the slope of change when $\theta_u \neq b_q$. In 1P-IRF, $a_q$ is 1 for all questions.

$$f(u, q) = \sigma(a_q(\theta_u - b_q)) = \frac{1}{1 + e^{-a_q(\theta_u - b_q)}} \tag{2}$$

IRT requires the following on student and question parameters:

1. student ability $\theta_u$ should not contain question level information, and question parameters $b_q$ and $a_q$ should not contain student information.

2. $\theta_u$ and $b_q$ should be on the same continuum so that they can be compared directly.

3. $a_q$ must be positive so that $f(u, q)$ increases with increased $\theta_u$ and decreased $b_q$.

4. When $\theta_u = b_q$, there is even odd of answering correctly or wrongly, i.e., $f(u, q)$=0.5.

Existing IRT-based models do not meet all of the above requirements which causes inconsistent outputs as shown in Figure 2. In the next section, we propose a model to satisfy all these requirements so that the final predictions are consistent with the learned IRT parameters and these outputs can be used easily for skill-meter building and other downstream tasks.

# 4 THE PROPOSED SKKT-IRT MODEL

In this section, we introduce our IRT-based deep knowledge tracing model called SKKT-IRT which learns skill-level student abilities without sacrificing accuracy or consistency. We first describe its overall architecture, and then describe individual components.

## 4.1 THE OVERALL ARCHITECTURE

The overall architecture of our model is shown in Figure 3. It consists of an LSTM sublayer to generate representations of student history sequences, a student ability network to map outputs of the LSTM sublayer and skill ID embeddings to student abilities over skills, a question difficulty network to transform query sequence embeddings to question difficulties, and a question discrimination network to transform query sequence embeddings to question discrimination parameters in 2P-IRF.

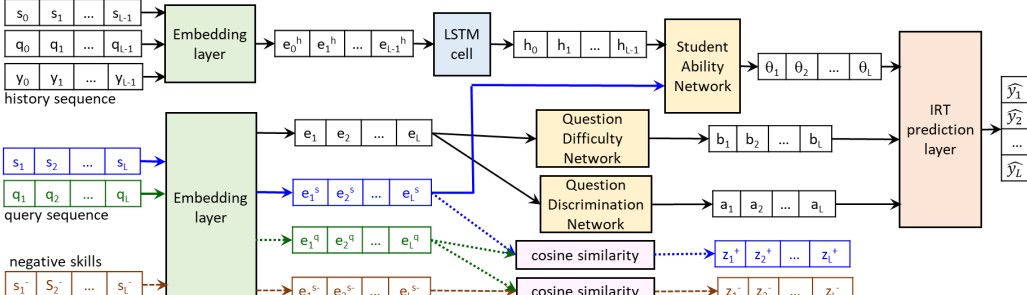

Figure 3: Architecture of SKKT-IRT. Length-$L$ history sequence is fed to LSTM to generate hidden representation of history sequence. Length-$L$ query embedding sequence is used to generate question difficulty and discrimination parameters. Query sequence is one position ahead of history sequence. $s_i$s are skill IDs, $q_i$s are question IDs, and $y_i$s are class labels at position $i$, $i$=0, 1, $\cdots$, $L$.

Our model takes length-$(L + 1)$ learning activity sequences as inputs. For a given length-$(L + 1)$ learning activity sequence, its first length-$L$ sub-sequence is regarded as *history sequence* and is fed to the LSTM sublayer, and the last length-$L$ sub-sequence is regarded as *query sequence* whose class labels are to be predicted. Note that the query sequence (from 1 to $L$) is one position ahead of the history sequence (from 0 to $L - 1$). History sequences contain skill IDs, question IDs and class labels. Query sequences contain skill IDs and question IDs only. The model is trained to predict the class labels over the whole length-$L$ query sequence.

## 4.2 EMBEDDING LAYER

The inputs to our model include skill IDs in $S$, question IDs in $Q$ and student responses $\in \{0, 1\}$ (class labels). Skill IDs are mapped to $d_s$-dimensional embeddings using an embedding matrix $M_S \in \mathcal{R}^{|S| \times d_s}$, where the $i$-th row vector of $M_S$ is the embedding of skill ID $i$ and it is $d_s$-dimensional. Question IDs are mapped to $d_q$-dimensional embeddings using an embedding matrix $M_Q \in \mathcal{R}^{|Q| \times d_q}$, where the $i$-th row vector of $M_Q$ is the $d_q$-dimensional embedding of question ID $i$ and $d_q$ can be different from $d_s$. Class labels are mapped to $d$-dimensional embeddings using an embedding matrix $\mathcal{M}_\mathcal{C} \in \mathcal{R}^{2 \times d}$, where $d$ is the input dimension to the LSTM sublayer and it can be different from $d_s$ and $d_q$.

We use $q_i$ to denote question ID, $s_i$ to denote skill ID, and $y_i$ to denote class label at position $i$ in a length-$(L + 1)$ sequence, $i$=0, 1, $\cdots$, $L$. To generate the input vectors to the LSTM sublayer on history sequences, skill ID embeddings and question ID embeddings are concatenated and then linearly transformed to $d$-dimensional vectors, and then added to the class label embeddings. More formally, let $e_i^s$ be the skill ID embedding, $e_i^q$ be the question ID embedding, and $e_i^y$ be the class label embedding at position $i$, $i$=0, 1, $\cdots$, $L - 1$. The input vector $e_i^h$ to LSTM is a $d$-dimensional vector and it is generated as follows, where $W_1 \in \mathcal{R}^{(d_s + d_q) \times d}$ and $d_1 \in \mathcal{R}^d$ are learnable parameters.

$$e_i^h = Dropout(([e_i^s, e_i^q] \cdot W_1 + d_1) + e_i^y), i = 0, 1, \cdots, L - 1 \tag{3}$$

Let $e_i^s$ be the skill ID embedding, and $e_i^q$ be the question ID embedding at position $i$ of the query sequence, $i=1, 2, \cdots, L$. The query sequence is converted to $d$-dimensional input vectors to the question difficulty network and the question discrimination network as follows.

$$e_i = Dropout([e_i^s, e_i^q] \cdot W_1 + d_1), i = 1, 2, \cdots, L \tag{4}$$

Note that skill ID embeddings, question ID embeddings, $W_1$ and $d_1$ are shared between history sequences and query sequences. If more than one skill IDs are associated with a question, the embeddings of these skill IDs are summed together and then the resultant $d_s$-dimensional vector is concatenated with the question ID embedding.

### 4.3 THE STUDENT ABILITY NETWORK

The LSTM sublayer takes $e_i^h$ generated by Equation 3 as inputs. The hidden state of the LSTM cell at position $i-1$, denoted as $h_{i-1}$, is regarded as the representation of the history sequence for query question $q_i$ at position $i$. The student ability network converts the concatenation of $h_{i-1}$ and skill ID embedding $e_i^s$ to skill-level student ability $\theta_i^s$ at position $i$. It is a two-layer feed-forward neural network (FNN) given as below, where $W_2 \in \mathcal{R}^{(d+d_s) \times d_f}$, $b_2 \in \mathcal{R}^{d_f}$, $W_3 \in \mathcal{R}^{d_f \times 1}$ and $b_3 \in \mathcal{R}$ are learnable model parameters, $d_f$ is the hidden dimension of the FNN network.

$$\theta_i^s = \sigma(Dropout(ReLU(BatchNorm([h_{i-1}, e_i^s]) \cdot W_2 + b_2)) \cdot W_3 + b_3) \tag{5}$$

A batch normalization layer is applied to the concatenation of $h_{i-1}$ and $e_i^s$ before FNN is applied. The value range of $\theta_i$s is controlled to be within (0, 1) using the sigmoid function. Question level information is not used to generate student abilities, so **requirement 1 of IRT is satisfied**.

### 4.4 THE QUESTION DIFFICULTY NETWORK

The question difficulty network maps query sequence embeddings $e_i$s generated by Equation 4 to question difficulties using a two-layer feed-forward neural network as follows, where $W_4 \in \mathcal{R}^{d \times d_f}$, $b_4 \in \mathcal{R}^{d_f}$, $W_5 \in \mathcal{R}^{d_f \times 1}$, $b_5 \in \mathcal{R}$ are learnable model parameters, and $d_f$ is the hidden dimension of the FNN network.

$$b_i = \sigma(Dropout(ReLU(BatchNorm(e_i) \cdot W_4 + b_4)) \cdot W_5 + b_5) \tag{6}$$

A batch normalization layer is applied to $e_i$ before FNN is applied. The value range of $b_i$s is also controlled using the sigmoid function to be within (0, 1) like that of student abilities. This ensures that **requirement 2 of IRT is satisfied**.

Question difficulties can also be estimated directly from training data. We use the same method as in (Liu et al., 2024) to estimate question difficulties as follows, where $n$ is the number of first attempts of $q$ by students in training data, $n_p$ be the number of activities with positive class labels among the $n$ activities, $p_1$ be the overall percentage of correct answers in training data, and $\lambda$ is used for smoothing and it is set to 5 in our experiments.

$$\hat{b}_q = 1 - \frac{n_p + \lambda * p_1}{n + \lambda} \tag{7}$$

We restrict that question difficulty $b_i$ learned using the question difficulty network should not deviate too much from that estimated using Equation 7, and a penalty is imposed if it deviates using L2 loss as follows. This is called question difficulty loss.

$$\mathcal{L}_b = \sqrt{\frac{\sum_{i=1}^{L}(b_i - \hat{b}_i)^2}{L}} \tag{8}$$

### 4.5 THE QUESTION DISCRIMINATION NETWORK

Question discrimination parameters control the slope of change when student ability $\theta_u$ and question difficulty $b_q$ are not equal. They must be positive numbers so that the predicted probability increases with increased $\theta_u$ and decreases with increased $b_q$. The question discrimination network maps query sequence embeddings to question discrimination parameters. It consists of a two-layer feed-forward

neural network and an activation function as defined in Equation 10, where $W_6 \in \mathcal{R}^{d \times d_f}$, $b_6 \in \mathcal{R}^{d_f}$, $W_7 \in \mathcal{R}^{d_f \times 1}$ and $b_7 \in \mathcal{R}$ are learnable model parameters.

$$a_i' = Dropout(ReLU(BatchNorm(e_i) \cdot W_6 + b_6)) \cdot W_7 + b_7 \tag{9}$$

$$a_i = \left\{ \begin{array}{ll} 1 + \log(1 + a_i') & a_i' \geq 0 \\ (1 + \log(1 - a_i'))^{-1} & a_i' < 0 \end{array} \right. \tag{10}$$

Question discrimination parameters are additional parameters used in 2P-IRF. In 1P-IRF, question discrimination is 1 for all questions. In Equation 10, when $a_i' = 0$, $a_i = 1$; when $a_i' > 0$, $a_i > 1$; when $a_i' < 0$, $0 < a_i < 1$. We choose to use $\log$ function in Equation 10 so that $a_i$ does not change too fast with the change of $a_i'$. The question discrimination parameters learned by Equation 10 are always positive, so **requirement 3 of IRT is satisfied**. To better align 1P-IRF and 2P-IRF, we add the following regularization term called question discrimination loss to ensure that question discrimination parameters are not too far away from 1 unless the deviation improves model accuracy.

$$\mathcal{L}_a = \sqrt{\frac{\sum_{i=1}^{L} (a_i'' - 1)^2}{L}}, a_i'' = \left\{ \begin{array}{ll} a_i & a_i > 1 \\ \frac{1}{a_i} & 0 < a_i < 1 \end{array} \right. \tag{11}$$

### 4.6 THE IRT PREDICTION LAYER

SKKT-IRF supports both 1P-IRF and 2P-IRF. For 1P-IRF, the question discrimination network, its outputs $a_i$s and $\mathcal{L}_a$ are not used. Class label at position $i$ is predicted as follows using 1P-IRF.

$$\hat{y}_i = \sigma(5 * (\theta_i^s - b_i)) \tag{12}$$

Similar to MIKT, we use a constant factor of 5 to extend the value range of $\hat{y}_i$ and it is multiplied to $(\theta_i - b_i)$. When $\theta_i = b_i$, $\hat{y}_i = \sigma(0) = 0.5$, so **requirement 4 of IRT is satisfied**.

Class label at position $i$ is predicted using 2P-IRF as follows:

$$\hat{y}_i = \sigma(a_i * 5 * (\theta_i^s - b_i)) \tag{13}$$

### 4.7 LEARNING QUESTION-SKILL RELATIONSHIPS AS AN AUXILIARY TASK

SKKT-IRT learns question-skill relationships as an auxiliary task so that the embedding of a skill can better capture the information of its questions. A question-skill pair is positive if the skill is associated with the question. In each batch, we randomly sample negative skills for questions, and then use the cosine similarity between the skill embeddings and question embeddings to predict whether the skills are associated with the questions as follows, where $e_i^q$ is the embedding of the question at position $i$, $e_i^s$ is the embedding of the positive skill at position $i$, and $e_i^{s-}$ is the embedding of the negative skill at position $i$.

$$\hat{z_i^+} = cosine\_similarity(e_i^q, e_i^s), i = 1, 2, \cdots, L \tag{14}$$

$$\hat{z_i^-} = cosine\_similarity(e_i^q, e_i^{s-}), i = 1, 2, \cdots, L \tag{15}$$

The relationship loss $\mathcal{L}_R$ is calculated using binary cross entropy loss as below:

$$\mathcal{L}_R = \frac{1}{2L} \sum_{i=1}^{L} (-\log(\hat{z_i^+}) - log(1 - \hat{z_i^-})) \tag{16}$$

Learning question-skill relationships has been explored in (Liu et al., 2020) to pre-train question and skill embeddings for knowledge tracing. Here we jointly optimize the relationship loss and the loss of the main task as described in the next subsection.

### 4.8 TRAINING LOSS

We use binary cross entropy loss $\mathcal{L}_{label}$, question difficulty loss $\mathcal{L}_b$, question discrimination loss $\mathcal{L}_a$ and question-skill relationship loss $\mathcal{L}_R$ to learn model parameters. Binary cross entropy loss

between the ground-truth class labels $y_i$s and predicted probabilities $\hat{y}_i$s over the whole length-$L$ query sequence is calculated below.

$$\mathcal{L}_{label} = \frac{1}{L} \sum_{i=1}^{L} (-y_i \log(\hat{y}_i) - (1 - y_i)log(1 - \hat{y}_i)) \quad (17)$$

The overall loss combines the four losses as below, where $\alpha$, $\beta$ and $\gamma$ are hyper-parameters.

$$\mathcal{L} = \mathcal{L}_{label} \ + \ \alpha\mathcal{L}_b + \beta\mathcal{L}_a + \gamma\mathcal{L}_R \quad (18)$$

## 5 EXPERIMENT RESULTS

In this section, we first introduce the datasets and settings used in our experiments, and then present the results of the following experiments: 1) comparing prediction accuracy and efficiency of SKKT-IRT with baseline deep KT models; 2) ablation studies; 3) showing distribution of learned IRT parameters by different IRT-based deep KT models to see whether they satisfy the four requirements of IRT; and 4) studying the correlation between question difficulties learned by different models with those estimated from data.

### 5.1 EXPERIMENT SETTINGS

The datasets used in our experiments and their statistics are listed in Table 1. For all the datasets, students with less than 10 activities are removed. The statistics are calculated after the removal. The last column is average sequence length. More details of the datasets can be found in Appendix A.

Table 1: Dataset statistics

| datasets | #students | #skills | #questions | #activities | % of corrects | avg_len |
|----------|-----------|---------|------------|-------------|---------------|---------|
| algebra05 | 571 | 138 | 52,846 | 813,632 | 76.7% | 1424.9 |
| assist09 | 3168 | 150 | 26,628 | 341,879 | 64.5% | 107.9 |
| assist17 | 1708 | 102 | 3,162 | 936,572 | 37.3% | 548.3 |
| ednet_10k | 10000 | 189 | 12,202 | 2,215,069 | 65.6% | 221.5 |

We include several groups of baseline models in our experiments:

- KT models using skills and responses only: DKT (Piech et al., 2015) and KQN (Lee & Yeung, 2019);
- KT models using questions and responses only: DKVMN (Zhang et al., 2017) and SAKT (Pandey & Karypis, 2019);
- KT models using skills, questions and responses: AKT (Ghosh et al., 2020), LPKT (Shen et al., 2021), IEKT (Long et al., 2021), DIMKT (Shen et al., 2022), simpleKT (Liu et al., 2023b) and QIKT (Chen et al., 2023);
- IRT-based deep KT models: Deep-IRT (Yeung, 2019), DKT-IRT (Converse et al., 2021), PKT (Sun et al., 2024a) and MIKT (Sun et al., 2024b).

More details on these baseline models and hyper-parameter tuning can be found in Appendix B and Appendix D.

### 5.2 COMPARISON WITH BASELINES

Table 2 shows the mean and standard deviation of model AUC evaluated using five-fold cross-validation. The last column is the mean AUC over all the datasets. Accuracy of the models are reported in Appendix F. The best performance is highlighted in **bold**. The second best performance is highlighted using underline. We include two variants of SKKT-IRT for comparison. They use 1P-IRF and 2P-IRF respectively at their prediction layer, and they are denoted as SKKT-IRT1 and SKKT-IRT2 respectively. MIKT has the highest mean AUC among all the baseline models. The mean AUC of our model is higher than all baseline models, though the gap between MIKT and our

Table 2: Comparison with baseline models. SKKT-IRT1 uses 1P-IRF. SKKT-IRT2 uses 2P-IRF.

| models | algebra05 | assist09 | assist17 | ednet_10k | mean |
|---|---|---|---|---|---|
| DKT | 0.6798±0.0081 | 0.7203±0.0043 | 0.7129±0.0111 | 0.6903±0.0075 | 0.7008 |
| KQN | 0.6853±0.0071 | 0.7302±0.0058 | 0.7232±0.0028 | 0.6861±0.0077 | 0.7062 |
| DKVMN | 0.7885±0.0035 | 0.7229±0.0048 | 0.7491±0.0029 | 0.7460±0.0052 | 0.7516 |
| SAKT | 0.8015±0.0029 | 0.7351±0.0045 | 0.7210±0.0065 | 0.7521±0.0048 | 0.7524 |
| AKT | 0.8166±0.0021 | 0.7857±0.0016 | 0.7795±0.0049 | 0.7601±0.0051 | 0.7855 |
| LPKT | 0.8083±0.0036 | 0.7586±0.0033 | 0.7939±0.0028 | 0.7573±0.0042 | 0.7795 |
| IEKT | 0.8164±0.0041 | 0.7728±0.0021 | 0.7862±0.0040 | 0.7449±0.0081 | 0.7801 |
| DIMKT | 0.8186±0.0023 | 0.7801±0.0016 | 0.7841±0.0020 | 0.7555±0.0054 | 0.7846 |
| simpleKT | 0.8163±0.0021 | 0.7790±0.0025 | 0.7780±0.0049 | 0.7558±0.0054 | 0.7823 |
| QIKT | 0.8135±0.0028 | 0.7018±0.0030 | 0.7810±0.0040 | 0.7512±0.0053 | 0.7619 |
| Deep-IRT | 0.7743±0.0039 | 0.7159±0.0058 | 0.7475±0.0025 | 0.7435±0.0043 | 0.7453 |
| DKT-IRT | 0.7842±0.0055 | 0.6970±0.0049 | 0.6985±0.0194 | 0.7119±0.0054 | 0.7229 |
| PKT | 0.7480±0.0052 | 0.6886±0.0068 | 0.6394±0.0087 | 0.7083±0.0052 | 0.6961 |
| MIKT | 0.8224±0.0023* | 0.7914±0.0020* | 0.7700±0.0080 | **0.7645**±0.0045 | 0.7871 |
| SKKT-IRT1 | 0.8197±0.0026 | **0.7923**±0.0015 | 0.7932±0.0041 | 0.7574±0.0052 | 0.7907 |
| SKKT-IRT2 | **0.8230**±0.0019 | 0.7920±0.0015 | **0.7962**±0.0037 | 0.7580±0.0052 | **0.7923** |

Table 3: Ablation studies.

| models | algebra05 | assist09 | assist17 | ednet_10k | mean |
|---|---|---|---|---|---|
| SKKT-IRT1-C | 0.8155±0.0022 | 0.7792±0.0030 | 0.7917±0.0023 | 0.7551±0.0052 | 0.7854 |
| SKKT-IRT2-C | 0.8133±0.0023 | 0.7737±0.0026 | 0.7939±0.0031 | 0.7578±0.0051 | 0.7847 |
| SKKT-IRT1-R | 0.8150±0.0027 | 0.7789±0.0018 | 0.7912±0.0031 | 0.7569±0.0048 | 0.7855 |
| SKKT-IRT2-R | 0.8137±0.0022 | 0.7766±0.0030 | 0.7962±0.0037 | 0.7580±0.0052 | 0.7861 |
| SKKT-IRT1 | 0.8197±0.0026 | 0.7923±0.0015 | 0.7932±0.0041 | 0.7574±0.0052 | 0.7907 |
| SKKT-IRT2 | 0.8230±0.0019 | 0.7920±0.0015 | 0.7962±0.0037 | 0.7580±0.0052 | 0.7923 |

model is small. We also studied the efficiency of all models. Our model is around 50 times faster than MIKT, about seven times faster than AKT and 14 times faster than DIMKT, whose AUC is the second and the third highest among all baselines respectively. More details of on the running time of the models can be found in Appendix E.

The mean AUC of the other three IRT-based deep KT models is all significantly lower than our model. The low AUC of Deep-IRT comes from its embedding layer which ignores skill IDs and its key-value memory network which is not as good as LSTM at capturing sequential information. The low AUC of DKT-IRT comes from its inefficient prediction layer where the output of LSTM is mapped to $K+1$-dimensional vectors, $K$ is the number of skills, and skill ID of the next question is used to get student ability. PKT suffers from the same problem as DKT-IRT in its MLP layers for generating student and question parameters.

## 5.3 ABLATION STUDIES

In this experiment, we study the effectiveness of the three techniques used in our model and the results are shown in Table 3. SKKT-IRT1-C and SKKT-IRT2-C use class label loss $\mathcal{L}_{label}$ only, and they do not use the other three losses. SKKT-IRT1-R and SKKT-IRT2-R use $\mathcal{L}_{label}$ and $\mathcal{L}_R$ only. Using the two regularization techniques can improve the AUC of SKKT-IRT2 by around 1.5% on *assist09* and by around 0.9% on algebra05, but they are not useful on the other two datasets. We recommend the use of the two regularization terms on datasets with a large number of questions. Using 2P-IRF improves the model performance very slightly than using 1P-IRF.

## 5.4 STATISTICS OF LEARNED IRT PARAMETERS

Table 4 shows the statistics of the IRF parameters generated by all IRT-based deep KT models. For DKT-IRT, the value range of its question difficulty parameters is quite different from that of student abilities, which violates requirement 2 of IRT. Also, question discrimination parameters generated

Table 4: Statistics of IRF parameters learned by different models.

| models | | algebra05 | | | assist09 | | | assist17 | | | ednet_10k | | |
|---|---|---|---|---|---|---|---|---|---|---|---|---|---|
| | | min | max | mean | min | max | mean | min | max | mean | min | max | mean |
| Deep-IRT | $\theta$ | -0.97 | 0.98 | 0.54 | -0.99 | 0.99 | 0.32 | -0.91 | 0.91 | -0.16 | -0.79 | 0.87 | 0.22 |
| | $b$ | -0.71 | 0.69 | -0.09 | -1.00 | 1.00 | -0.08 | **-0.57** | **0.58** | -0.01 | -0.71 | 0.64 | -0.10 |
| MIKT | $\theta$ | 0.01 | 1.00 | 0.80 | 0.00 | 1.00 | 0.66 | 0.00 | 1.00 | 0.36 | 0.00 | 1.00 | 0.66 |
| | $b$ | **0.36** | **0.65** | 0.48 | **0.40** | **0.61** | 0.49 | 0.07 | 0.91 | 0.47 | **0.22** | **0.78** | 0.48 |
| SKKT-IRT1 | $\theta$ | 0.00 | 0.93 | 0.54 | 0.00 | 0.96 | 0.52 | 0.00 | 1.00 | 0.48 | 0.01 | 0.98 | 0.52 |
| | $b$ | 0.00 | 0.97 | 0.23 | 0.00 | 0.91 | 0.34 | 0.00 | 1.00 | 0.57 | 0.00 | 0.98 | 0.32 |
| DKT-IRT | $\theta$ | -22.18 | 21.65 | 0.29 | -20.47 | 20.80 | 3.18 | -11.47 | 25.40 | -0.22 | -21.64 | 23.31 | -0.10 |
| | $b$ | -0.13 | 0.13 | -0.04 | -0.13 | 0.12 | -0.03 | -0.70 | 0.78 | 0.07 | -0.53 | 0.41 | -0.04 |
| | $a$ | -0.32 | 0.31 | -0.01 | -0.25 | 0.24 | 0.00 | -1.50 | 1.50 | 0.00 | -1.26 | 1.34 | 0.02 |
| PKT | $\theta$ | **0.50** | 1.00 | 0.97 | 0.00 | 1.00 | 0.79 | **0.32** | **0.70** | 0.50 | 0.00 | 1.00 | 0.86 |
| | $b$ | 0.00 | 1.00 | 0.35 | 0.00 | 1.00 | 0.44 | 0.06 | 0.99 | 0.55 | 0.00 | 1.00 | 0.43 |
| | $a$ | 0.00 | 1.00 | 0.62 | 0.00 | 1.00 | 0.58 | 0.02 | 0.99 | 0.46 | 0.00 | 1.00 | 0.59 |
| SKKT-IRT2 | $\theta$ | 0.01 | 0.87 | 0.51 | 0.00 | 0.96 | 0.51 | 0.00 | 1.00 | 0.47 | 0.02 | 0.97 | 0.53 |
| | $b$ | 0.00 | 0.96 | 0.23 | 0.00 | 0.92 | 0.34 | 0.00 | 1.00 | 0.55 | 0.00 | 0.98 | 0.33 |
| | $a$ | 0.81 | 1.79 | 1.19 | 0.99 | 1.02 | 1.00 | 0.99 | 1.02 | 1.00 | 0.97 | 1.03 | 1.00 |

Table 5: Pearson correlation coefficient between question difficulties learned by models and $\hat{b}$.

| | AKT | DKT-IRT | DeepIRT | PKT | MIKT | SKKT-IRT1 | SKKT-IRT2 |
|---|---|---|---|---|---|---|---|
| algebra05 | -0.085 | 0.940 | 0.602 | 0.423 | 0.919 | **0.999** | 0.996 |
| assist09 | -0.093 | 0.938 | 0.419 | 0.627 | 0.833 | **0.999** | **0.999** |
| assist17 | 0.250 | 0.832 | 0.770 | 0.417 | 0.781 | 0.866 | **0.936** |
| ednet_10k | 0.008 | 0.820 | 0.878 | 0.526 | 0.841 | **0.984** | 0.964 |
| mean | 0.020 | 0.882 | 0.667 | 0.498 | 0.843 | 0.962 | **0.974** |

by DKT-IRT can be negative, which violates requirement 3 of IRT. Even though Deep-IRT, PKT and MIKT restrict that student ability and question difficulty to be in the same range, but on some datasets, the value ranges of the two parameters can be different. These cases are highlighted in bold. PKT restricts the value range of question discrimination to be within [0, 1], while in reality, it can be larger than 1. The value ranges of the IRF parameters generated by our model conform very well to IRT. More specifically, student abilities and question difficulties have the same value range, and question discrimination parameters are positive numbers centered around 1.

## 5.5 CORRELATION BETWEEN IRT PARAMETERS LEARNED BY MODELS AND THOSE ESTIMATED FROM DATA

Table 5 shows Pearson correlation coefficients between question difficulties learned by different models and $\hat{b}$ calculated using Equation 7. AKT uses Rasch model (1P-IRF) at its embedding layer, so it also has question difficulty parameters and is included in Table 5. All the correlations are statistically significant with p-value $< 0.05$ except AKT on dataset *ednet_10k*. Question difficulty parameters learned by AKT have substantially weaker correlations with $\hat{b}$ than other models. PKT has the second weakest correlations. The question difficulties learned by our model have the strongest correlation with $\hat{b}$ due to the use of the question difficulty loss $\mathcal{L}_b$.

## 6 SUMMARY AND CONCLUSION

In this paper, we propose an IRT-based deep KT model which learns skill-level student abilities with SOTA accuracy and consistent outputs. The skill-level student abilities and other IRT parameters generated by our model can be easily used for skill-meter building and other downstream tasks like adaptive learning and personalized recommendations. Our model is also very efficient. The question difficulties learned by our model has higher correlation with those estimated from data than existing IRT-based deep KT models.

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

## A  DATA PROCESSING

**algebra05** (Stamper et al., 2010)[1] was used for KDD Cup 2010 Educational Data Mining Challenge. On this dataset, values of column "Problem Hierarchy" are used as skills, and combinations of values in "Problem Name" column and "Step Name" column are used as questions. All values are converted to lower case. We also replace concrete numbers in "Step Name" by variable names like $a$, $b$, $c$ so that similar step names can be merged together and regarded as the same step.

**assist09** [2] was collected on the ASSISTments platform in the school year of 2009-2010. We use the skill builder dataset. On this dataset, a question may have more than one skills, and we map combinations of skill IDs to single skill IDs.

**assist17** [3] was also collected on the ASSISTments platform and used in ASSISTments Data Mining Competition 2017. It contains student responses to math questions across two academic years.

**ednet_10k** [4] was collected by Santa—a multi-platform tutor for English learning. The original dataset is very large. We sampled 10000 students to form a smaller dataset. On this dataset, one question can have up to six skills (tags). For DLKT models that do not allow multiple skills per question, we map combinations of skill IDs to single skill IDs and there are 1482 unique combinations in the sampled data.

## B  BASELINE MODELS

Table 6 shows the base deep learning models used by baseline deep KT models, which IRF function they use, and whether skill IDs and question IDs are used.

For DKVMN, DIMKT, simpleKT, KQN, QIKT and Deep-IRT, we obtain their model implementations from the pyKT library (Liu et al., 2022). The implementations of AKT, LPKT, IEKT, PKT and IEKT are downloaded from the links provided in the original paper. We implemented DKT, SAKT and DKT-IRT ourselves based on their original papers.

## C  TRAINING AND TESTING

All the models use the same data loader and training and testing process. During the training phase, sequences are sampled from students' full learning activity sequences randomly. In each epoch, students with more activities are sampled more frequently. More specifically, the frequency that a student $u$ is sampled in each epoch is calculated as $\lceil N_u/(L+1) \rceil$, where $N_u$ is the number of activities of student $u$ and $L$ is the length of the sequences to be fed to knowledge tracing models. Once a student is sampled, a random position from this student's full activity sequence is picked as the ending position of the sampled segment. Using this sampling method, for a same student, different segments are sampled from this student's full activity sequence in different epochs, which has some regularization effect on model performance. All the sampled sequences have length $L+1$. Sampled

---

[1] https://pslcdatashop.web.cmu.edu/KDDCup/

[2] https://sites.google.com/site/assistmentsdata/home/2009-2010-assistment-data

[3] https://sites.google.com/view/assistmentsdatamining/dataset

[4] https://github.com/riiid/ednet

Table 6: Baseline models

| models | deep model | IRF | skill ID | ques ID |
|---|---|---|---|---|
| DKT (Piech et al., 2015) | LSTM | - | yes | no |
| KQN (Lee & Yeung, 2019) | LSTM | - | yes | no |
| DKVMN (Zhang et al., 2017) | memory | - | no | yes |
| SAKT (Pandey & Karypis, 2019) | attention | - | no | yes |
| AKT (Ghosh et al., 2020) | attention | - | yes | yes |
| LPKT (Shen et al., 2021) | sequential | - | Q-matrix | yes |
| IEKT (Long et al., 2021) | sequential | - | yes | yes |
| DIMKT (Shen et al., 2022) | sequential | - | yes | yes |
| simpleKT (Liu et al., 2023b) | attention | - | yes | yes |
| QIKT (Chen et al., 2023) | LSTM | - | yes | yes |
| Deep-IRT (Yeung, 2019) | memory | 1P-IRF | no | yes |
| DKT-IRT (Converse et al., 2021) | LSTM | 2P-IRF | Q-matrix | yes |
| PKT (Sun et al., 2024a) | LSTM | 2P-IRF | yes | yes |
| MIKT (Sun et al., 2024b) | sequential+graph | 1P-IRF | yes | yes |

sequences with length less than $L+1$ are padded with zeros at the beginning of the sequences. During the inference phase, every testing activity $x$ is used as the last activity of a sequence, and the $L$ activities prior to $x$ are used to form a length-($L+1$) testing sequence to be passed to knowledge tracing models.

## D  HYPER-PARAMETER TUNING

On all datasets, the following fixed hyper-parameters are used for all models (if applicable) so that all models have comparable size: skill embedding dimension $d_s$ and model input dimension $d$ are both set to 64, hidden layer dimension of FNN is set to 512, number of RNN and attention layers is set to two, and attention head number is set to eight.

Grid search is used to select the best values for the following hyper-parameter values on validation data for all models. The maximum learning rate is selected from [0.01, 0.003, 0.001, 0.0003, 0.0001]. Dropout rate is selected from [0, 0.1, 0.2, 0.3, 0.4, 0.5]. For SKKT-IRF, $\alpha$, $\beta$ and $\gamma$ are tuned using values from [0, 0.03, 0.1, 0.3, 0.5, 1]. Number of latent concepts for DKVMN and Deep-IRT is selected from [4, 8, 16, 32, 64]. The number of questions on *assist09* and *algebra05* is very large. To avoid over-parameterization, we tune the dimension of question ID embeddings for applicable models using values from [1, 2, 4, 8, 16, 32, 64].

For training, sequence length $L$ is set to 200 and batch size is set to 256. Adam optimizer is used for model training. All models are trained using one cycle of cosine annealing scheduling with a minimum learning rate of 0.0001, and the number of epochs is set to 100. Early stopping is used if the performance of a model does not improve after 20 epochs.

## E  RUNNING TIME

Table 7 shows the time requried for training one epoch by different models. This experiment was conducted on an NVIDIA A40 GPU with 48GB memory. The last column is the ratio of the epoch time of a model to the epoch time of SKKT-IRT2. MIKT, AKT and DIMKT are the top-3 baselines with the highest AUC. They are 51.7, 7.5 and 13.9 times slower than our model.

## F  ACCURACY OF MODELS

Table 8 shows the accuracy of our model and baseline models.

Paired t-test over five folds is used to get the statistical significance of the improvement achieved by our model. For AUC reported in Table 2 and accuracy reported in Table 8, if the performance of our model is higher than that of a baseline model, the improvement is always statistically significant with p-value $< 0.05$ except for the cases marked by "*".

Table 7: Time for training one epoch by different models in seconds.

| models | algebra05 | assist09 | assist17 | ednet_10k | mean | x |
|---|---|---|---|---|---|---|
| DKT | 0.8 | 0.7 | 0.9 | 2.1 | 1.1 | 0.6 |
| KQN | 0.9 | 0.8 | 1.0 | 2.8 | 1.4 | 0.7 |
| DKVMN | 5.8 | 3.7 | 7.3 | 17.0 | 8.4 | 4.6 |
| SAKT | 1.0 | 0.9 | 1.2 | 3.0 | 1.5 | 0.8 |
| AKT | 7.6 | 7.0 | 9.5 | 31.1 | 13.8 | **7.5** |
| LPKT | 21.6 | 21.4 | 27.0 | 103.8 | 43.4 | 23.5 |
| IEKT | 33.8 | 31.5 | 35.9 | 111.5 | 53.2 | 28.7 |
| DIMKT | 13.0 | 14.9 | 17.0 | 58.2 | 25.8 | **13.9** |
| simpleKT | 2.6 | 2.5 | 3.3 | 9.5 | 4.5 | 2.4 |
| QIKT | 9.2 | 4.8 | 2.0 | 13.9 | 7.5 | 4.0 |
| Deep-IRT | 3.9 | 3.8 | 5.1 | 15.1 | 7.0 | 3.8 |
| DKT-IRT | 1.1 | 1.0 | 1.3 | 3.1 | 1.6 | 0.9 |
| PKT | 1.5 | 1.3 | 1.7 | 14.8 | 4.8 | 2.6 |
| MIKT | 46.5 | 45.0 | 57.9 | 233.3 | 95.7 | **51.7** |
| SKKT-IRT1 | 1.0 | 0.9 | 1.2 | 3.1 | 1.5 | 0.8 |
| SKKT-IRT2 | 1.2 | 1.0 | 1.4 | 3.7 | 1.9 | 1.0 |

Table 8: Comparison with baseline models on Accuracy.

| models | algebra05 | assist09 | assist17 | ednet_10k | mean |
|---|---|---|---|---|---|
| DKT | $0.7733\pm0.0104$ | $0.6966\pm0.0133$ | $0.6844\pm0.0024$ | $0.6887\pm0.0065$ | 0.7107 |
| KQN | $0.7728\pm0.0104$ | $0.7106\pm0.0034$ | $0.6882\pm0.0040$ | $0.6864\pm0.0069$ | 0.7145 |
| DKVMN | $0.8041\pm0.0078$ | $0.6955\pm0.0073$ | $0.7050\pm0.0030$ | $0.7156\pm0.0062$ | 0.7301 |
| SAKT | $0.8103\pm0.0075$ | $0.6980\pm0.0067$ | $0.6932\pm0.0055$ | $0.7178\pm0.0054$ | 0.7298 |
| AKT | $0.8181\pm0.0074$ | $0.7413\pm0.0031$ | $0.7229\pm0.0032$ | $0.7237\pm0.0056$ | 0.7515 |
| LPKT | $0.8137\pm0.0078$ | $0.7258\pm0.0051$ | $\underline{0.7356\pm0.0040}$* | $0.7219\pm0.0053$ | 0.7492 |
| IEKT | $0.8167\pm0.0065$ | $0.7302\pm0.0033$ | $0.7280\pm0.0042$ | $\underline{0.7246\pm0.0094}$ | 0.7499 |
| DIMKT | $0.8178\pm0.0067$ | $0.7381\pm0.0048$ | $0.7267\pm0.0018$ | $0.7203\pm0.0058$ | 0.7507 |
| simpleKT | $0.8166\pm0.0065$ | $0.7347\pm0.0033$ | $0.7210\pm0.0007$ | $0.7205\pm0.0058$ | 0.7482 |
| QIKT | $0.8180\pm0.0073$ | $0.6733\pm0.0043$ | $0.7241\pm0.0026$ | $0.7170\pm0.0063$ | 0.7331 |
| Deep-IRT | $0.7954\pm0.0056$ | $0.6908\pm0.0043$ | $0.7010\pm0.0024$ | $0.7146\pm0.0056$ | 0.7255 |
| DKT-IRT | $0.7972\pm0.0081$ | $0.6796\pm0.0071$ | $0.6684\pm0.0035$ | $0.6988\pm0.0052$ | 0.7110 |
| PKT | $0.7715\pm0.0130$ | $0.6759\pm0.0081$ | $0.6313\pm0.0079$ | $0.6916\pm0.0049$ | 0.6926 |
| MIKT | $\mathbf{0.8206}\pm0.0069$ | $0.7449\pm0.0029$* | $0.7195\pm0.0028$ | $\mathbf{0.7265}\pm\mathbf{0.0053}$ | 0.7529 |
| SKKT-IRT1 | $0.8196\pm0.0076$ | $\mathbf{0.7456}\pm0.0033$ | $0.7330\pm0.0045$ | $0.7216\pm0.0062$ | $\underline{0.7550}$ |
| SKKT-IRT2 | $\mathbf{0.8206}\pm0.0073$ | $\underline{0.7450}\pm0.0036$ | $\mathbf{0.7361}\pm0.0036$ | $0.7221\pm0.0057$ | **0.7560** |

# G  LIMITATIONS AND FUTURE WORK

In this paper, we consider only question IDs, skill IDs and class labels. Other information in student learning activity data like timestamp, response time are not used. We will explore how to use such information effectively to model student learning and forgetting in our future work.

