# OpenReview forum: "Learning Skill-level Student Abilities with Item Response Theory"
_ICLR.cc/2025/Conference — Submitted to ICLR 2025_

### Official Review · Reviewer_ZDRM · 2024-10-30

**Soundness:** 2
**Presentation:** 1
**Contribution:** 2
**Rating:** 3
**Confidence:** 4

**Summary:**

This work proposes the SKKT-IRT model for knowledge tracing by combining Item Response Theory and neural networks. The model estimates skill-level student abilities with high accuracy and consistency. It addresses limitations in previous IRT-based models by ensuring outputs comply with IRT principles and using auxiliary tasks to improve performance while keeping computationally efficient.

**Strengths:**

The authors thoroughly summarized existing IRT-based approaches in knowledge tracing and identified key limitations, particularly highlighting the contradictions that arise in terms of interpretability when parameters are inferred through deep learning frameworks. To address these limitations, the authors proposed regularization methods that ensure the integration of neural networks enhances performance accuracy without undermining the principles of IRT.

The authors present a wide range of benchmarks, from classic knowledge tracing models to recent IRT- and neural network-based methods. The evaluation demonstrates the advantages of the proposed SKKT-IRT model in predicting student’s performance.

I also truly appreciate that visual aids are particularly effective. Figures 2 and 3 stand out as clear and intuitive, allowing readers to easily grasp the key concepts of IRT and the proposed architecture.

**Weaknesses:**

Writing: the writing needs substantial improvement for better flow and clarity. Key issues include 1) The abstract is unclear and fails to highlight the main motivations and contributions. Phrases like "a question difficulty network for learning question difficulties" are redundant; 2) The introduction provides too much detail about IRT without first explaining the model. For example, the discussion about the "question discrimination parameter" (lines 89-91) lacks necessary context, making it hard to understand the motivation; 3) While the methodology section is detailed, the formulas (Eq. 3-6) need more intuitive explanations and better formatting. Instead of presenting formulas which only show the network architecture, the rationale behind design choices should be clarified. Also, the notations need more structured explanations (duplicates in line 266 and 270).

Position of the paper: While the authors acknowledge existing IRT-based models in knowledge tracing, this work should also position itself among other interpretable KT models. I recommend providing more detailed comparisons in the literature review[1,2], particularly explaining why the focus is on IRT. Additionally, as noted in the limitations, the model does not account for student abilities, which is a crucial aspect of personalized education—especially given that the work claims such an application.

Understanding the IRF parameters: two of the contributions are learning skill-level student abilities and consistent IRT outputs. But the claim of student skill-level abilities should be demonstrated while in the current experiments there is no explanation. Also, the statistics (mean, min, max) of the question-level parameters are not convincing.

1) Regarding the question-level parameters:
-	I do not follow the purpose of Table 4 and the bold numbers. Why do authors claim that the student abilities and the question difficulties are not in the same range?
-	It is not informative when authors aggregate over all students if you want to do personalized education. I am curious about individual student’s abilities and the difficulties of the questions they encountered.
-	In Table 5, where do the ground-truth difficulties come from? Also, since the model regularizes the inferred parameters to some prior knowledge (Eq. 8 and 11). My questions are two folds:
    - How would the ablation study looks like if one relagurizer is removed at once? Also, what would be the experiment results if the empirical estimations $\hat{b}_q$ and $a = 1$ are used?
    - How would the empirical difficulty $\hat{b}_q$ correlates with the ground-truth? If it is correlated well, then the inferred difficulty parameters are not that useful.

2) Regarding the skill-level student abilities:
There is no analysis of these inferred latent variables for each student and each skill. The SOTA performance in predicting performance can not claim “The skill-level student abilities and other IRT parameters generated by our model can be easily used for skill-meter building and other downstream tasks like adaptive learning and personalized recommendations. “ I recommend authors provide statistical analysis for the inferred student abilities 1) are useful for personalized recommendations. E.g., show the inferred parameters are specific to students [3]; 2) are truly skill-level abilities. e.g., correlation between these ability parameters and held-out student performance.

**Questions:**

Major

-	Would it be possible to address the issues listed above among weaknesses?
-	The QIKT [4] model, a neural network based on IRT, is misclassified in Table 2. Experimental results for the assist09 dataset (0.7081 in this study versus 0.7878 reported in QIKT Table 2) show quite a lot discrepancies. Were different experimental settings used? Beyond skill-level prediction and regularization of the IRF parameters, what unique contributions distinguish SKK-IRT from QIKT?

Minor

-	In line 291, it states, “Question-level information is not used to generate student abilities.” However, question embeddings are used to train and inform the hidden state of the LSTM cell, $h$, which, in turn, influences the generation of student abilities. Could you clarify the basis of this claim?
-	There are several KT datasets. Why do you choose the four among others?



[1] Cui, J., Yu, M., Jiang, B., Zhou, A., Wang, J., & Zhang, W. (2024, May). Interpretable Knowledge Tracing via Response Influence-based Counterfactual Reasoning. In 2024 IEEE 40th International Conference on Data Engineering (ICDE) (pp. 1103-1116). IEEE.

[2] Cui, C., Ma, H., Zhang, C., Zhang, C., Yao, Y., Chen, M., & Ma, Y. (2023). Do We Fully Understand Students' Knowledge States? Identifying and Mitigating Answer Bias in Knowledge Tracing. arXiv preprint arXiv:2308.07779.

[3] Zhou, H., Bamler, R., Wu, C. M., & Tejero-Cantero, Á. (2024). Predictive, scalable and interpretable knowledge tracing on structured domains. arXiv preprint arXiv:2403.13179.

[4] Chen, J., Liu, Z., Huang, S., Liu, Q., & Luo, W. (2023, June). Improving interpretability of deep sequential knowledge tracing models with question-centric cognitive representations. In Proceedings of the AAAI Conference on Artificial Intelligence (Vol. 37, No. 12, pp. 14196-14204).

---

### Official Review · Reviewer_XnHS · 2024-11-02

**Soundness:** 3
**Presentation:** 2
**Contribution:** 3
**Rating:** 5
**Confidence:** 4

**Summary:**

The authors focus on skill-level knowledge state modeling and propose an LSTM-based knowledge tracing model, SKKT-IRT. The SKKT-IRT model uses LSTM to capture students' historical states, along with separate networks for estimating student ability, question difficulty, and question discrimination at the skill level. The model further incorporates a student ability network, question difficulty network, and question discrimination network to generate student and question parameters, using regularization techniques to ensure output consistency. Additionally, SKKT-IRT integrates an auxiliary task to learn question-skill relationships, enhancing the representation quality of skills by embedding information from relevant questions.Experimental results indicate that SKKT-IRT achieves state-of-the-art accuracy at the skill level and outperforms baseline models in both prediction accuracy and computational efficiency. Compared to other IRT-based deep KT models, SKKT-IRT demonstrates robust performance, producing outputs suitable for downstream applications such as adaptive learning and personalized recommendations.

Contributions:
1. Proposing an LSTM-based knowledge tracing model, SKKT-IRT, to estimate skill-level student abilities.
2. Introducing a model architecture that combines skill embeddings with student ability, question difficulty, and question discrimination networks, alongside regularization techniques for consistent output.
3. Validating SKKT-IRT's high accuracy, efficiency, and reliability through comparative experiments on diverse datasets.

**Strengths:**

The research motivation is clear: The paper proposes a skill-level modeling approach to address the limitations of existing IRT-KT models, which tend to be either too fine-grained at the problem level or generalized in overall ability assessment. This approach is well-motivated and aligns with the need to analyze students’ specific skill mastery in educational applications.

Use of auxiliary tasks to strengthen skill-problem associations: The paper captures the relationship between problems and skills through auxiliary tasks, enhancing the expressiveness of skill embeddings to better represent information from related problems.

The experimental design is comprehensive: The paper validates the model on four datasets and includes extensive comparisons with multiple baseline models, demonstrating the effectiveness of SKKT-IRT in different scenarios. Ablation experiments evaluate the effects of auxiliary tasks and regularization terms, providing support for the effectiveness of model components.

**Weaknesses:**

Lack of specific comparison and explanation for choosing LSTM: In the model’s embedding layer, the authors use LSTM to model students' historical sequences but do not explain the reason for this choice or whether they considered other potential architectures, such as Transformers or CNNs, as alternatives. While LSTM performs well for sequence modeling, it may have efficiency issues with long sequences or large-scale data. To strengthen the justification for their design, the authors could include comparisons between LSTM and other sequence models (such as Transformers or CNNs) in the experiments.

Lack of validation for the effectiveness of skill-level modeling: Although the authors propose an innovative approach to model student knowledge states at the skill level, the experimental validation primarily focuses on overall performance and parameter consistency. There is insufficient validation of the actual impact of skill-level modeling itself. The ablation study shows only the effect of different loss terms on the overall AUC, without providing specific insights into how well the model captures student abilities at the skill level. To further evaluate the model’s effectiveness, the authors could consider including a fine-grained analysis, such as examining the distribution of student abilities across different skills and their dynamic changes over time.

Limited discussion on the independent effects of the auxiliary task and regularization terms: The authors introduce a relationship loss as an auxiliary task to capture the relationships between questions and skills, enhancing skill embeddings to better represent question information. However, the importance and necessity of this auxiliary task lack theoretical explanation. In the ablation study, the authors explore the impact of the relationship loss and regularization terms on the model’s overall AUC, but this evaluation focuses on general effects across different datasets without analyzing the independent contributions of each loss term at the skill level or regarding IRT parameter consistency. The authors could conduct more detailed experiments by selectively removing or retaining each loss term, especially providing finer-grained comparisons at the skill level and in terms of IRT parameter alignment.

**Questions:**

Question1: What is the rationale for choosing LSTM to model student sequences in the model?

Question2: Are there any comparative experiments with Transformer or CNN that demonstrate LSTM’s suitability for this task?

Question3: Could you provide more details on how the model captures student knowledge states at the skill level? For example, could you show the distribution of student abilities across different skills and their dynamic changes over time to better understand the model's actual effectiveness at this level?

Question4: While relationship loss LR and regularization terms (Lb and La) impact overall AUC, their independent contributions at the skill level or to IRT parameter consistency remain unclear. Could you provide further evaluation of these loss terms' effects specifically at the skill level and regarding IRT consistency?

---

### Official Review · Reviewer_vMxd · 2024-11-03

**Soundness:** 3
**Presentation:** 3
**Contribution:** 2
**Rating:** 3
**Confidence:** 4

**Summary:**

The authors propose SKKT-IRT which estimates skills at KC level instead of a general student ability.

**Strengths:**

The paper is well written, the figures are explanatory.

**Weaknesses:**

Some requirements of IRT do not make sense to me. "$\theta$ and $b_q$ should be on the same continuum" Is it really a requirement? Because both are scalar anyway so both of them are in $\mathbb{R}$. Similarly, is 4 really a requirement?

Perhaps the authors should compare to psiKT who was presented at ICLR 2024.

> Zhou, Hanqi, et al. "Predictive, scalable and interpretable knowledge tracing on structured domains." The Twelfth International Conference on Learning Representations.

The proposed approach SKKT-IRT is very close to this other work, Section 3.2.3.

> Vie, Jill-Jênn, and Hisashi Kashima. "Deep Knowledge Tracing is an implicit dynamic multidimensional item response theory model." ICCE 2023-The 31st International Conference on Computers in Education. 2023.

Wilson et al. (2016) managed to match the performance of DKT with retrained IRT, which is clearly a very simple model.

> Wilson, Kevin H., et al. "Back to the basics: Bayesian extensions of IRT outperform neural networks for proficiency estimation." in Educational Data Mining 2016: 539.

If I compare to Gervet et al. 2020, I see that a logistic regression (Best LR) has a AUC of 0.831 on Algebra 2015 which outperforms the proposed approach SKKT-IRT.

> Gervet, Theophile, et al. "When is deep learning the best approach to knowledge tracing?." Journal of Educational Data Mining 12.3 (2020): 31-54.

According to this other paper, the performance of IKT on Assistments 2009 is 0.797 (which would outperform SKKT-IRT) and its performance on Algebra 2005 is 0.851. According to this same paper the performance of AKT on Algebra 2005 is 0.845 which would already outperform ExecCAKT.

> Minn, Sein, et al. "Interpretable knowledge tracing: Simple and efficient student modeling with causal relations." Proceedings of the AAAI conference on artificial intelligence. Vol. 36. No. 11. 2022.

I also encourage the authors to read about logistic knowledge tracing.

> Pavlik Jr, Philip Irvin, and Luke G. Eglington. "Automated search improves logistic knowledge tracing, surpassing deep learning in accuracy and explainability." Journal of Educational Data Mining 15.3 (2023): 58-86.

**Questions:**

Why do you limit your baselines to deep learning models while there is a large body of research (see Weaknesses) that shows that logistic-based approaches are simpler, more interpretable, and sometimes more accurate than their deep learning counterparts?

---

### Official Review · Reviewer_9RTX · 2024-11-05

**Soundness:** 2
**Presentation:** 2
**Contribution:** 1
**Rating:** 3
**Confidence:** 5

**Summary:**

This paper addresses a gap in knowledge tracing (KT) by proposing a model that estimates student knowledge at the skill level—essential for precise interventions but not effectively supported by existing methods. Current KT models either lack accuracy or offer knowledge estimates that are too broad or overly granular, making them impractical for teachers. The authors proposed the SKKT-IRT model integrates IRT principles with deep learning to provide accurate, skill-level insights while maintaining interpretability, also demonstrating the effective results in adaptive learning and personalized recommendations.

**Strengths:**

1. The content is relatively complete.
2. The paper compares a wider range of baselines to validate the effect and uses four datasets for the experiments.

**Weaknesses:**

**I don't think it's a very good paper especially on a topic related to knowledge tracing, and at least not up to what I would consider to be the standard of average ICLR acceptance.**

1. The motivation of this paper is not novel, and work on exploring the knowledge tracing task from the perspective of modeling skill dimensions has become common.

2. The writing of this paper does not flow well, for example, the Abstract section is too cumbersome, and in fact the description from motivation to challenge is too trivial not cohesive enough and not logical.

3. The design of the model is also very simple, simply adding an IRT interaction layer to the KT model that uses LSTM as a base module, and I don't see any novel design to solve the problems mentioned in the motivation.

4. The authors did not do interesting case studies to show such interpretable results as well as analysis, which is what makes the paper unattractive.

**Questions:**

Refer to the Weakness section.

---

> ### Comment · Reviewer_9RTX · 2024-12-02
>
> Since the author did not engage in the rebuttal, I am keeping my score.

---

### Meta-Review · Area_Chair_kn5h · 2024-12-11

**Metareview:**

This paper addresses an important problem: estimating student abilities at the skill level, which has great potential for adaptive learning and personalized recommendations. The proposed SKKT-IRT model combines Item Response Theory with deep learning, but its contributions are incremental and do not offer much novelty compared to existing methods. The use of LSTM for sequence modeling is not well-justified, and the authors did not explore alternatives like Transformers or simpler logistic-based methods which provide better interpretability or efficiency. Additionally, the paper does not thoroughly analyze the inferred skill-level parameters or how they could be used in practical applications, which weakens the clarity of its contributions. While the experiments show competitive performance, the absence of case studies and detailed validation of skill-level abilities reduces the overall impact of the work.

**Additional Comments On Reviewer Discussion:**

During the review period, reviewers pointed out concerns about the limited novelty of the method, the lack of exploration of alternative approaches, and insufficient analysis of skill-level modeling. However, the authors did not respond with rebuttals or additional clarifications.

---

### Decision · Program_Chairs · 2025-01-22

Reject